# The Role of Kiwifruit in Supporting Psychological Well-Being: A Rapid Review of the Literature

**DOI:** 10.3390/nu14214657

**Published:** 2022-11-03

**Authors:** Michael Billows, Naomi Kakoschke, Ian T. Zajac

**Affiliations:** 1Human Health, Health and Biosecurity, CSIRO, Adelaide 5000, Australia; 2School of Psychology, University of Adelaide, Adelaide 5000, Australia

**Keywords:** kiwifruit, mood, psychological well-being, vitamin C status

## Abstract

Consumption of vitamin-rich fruits and vegetables is emerging as a recommendation for the prevention and treatment of depression and anxiety. This review sought to examine literature investigating the role of kiwifruit in supporting psychological well-being in adult populations through increased vitamin C intake. The literature search using CINAHL, Embase and PubMed databases was restricted to English-language articles published from 2005 through July 2022. Inclusion criteria were randomized trials that delivered kiwifruit interventions to adult populations assessing psychological well-being. Studies were assessed for bias using the Joanna Briggs Institute critical appraisal tool for randomized controlled trials. The literature search identified two eligible trials involving 202 participants that delivered gold kiwifruit interventions and evaluated aspects of psychological well-being (e.g., mood disturbance, vitality, vigour, depression). Daily consumption of two gold kiwifruit was associated with significant reductions in mood disturbance and fatigue, and significant increases in well-being and vigour. Larger effects were observed in participants with higher baseline mood disturbance. Additional research involving a broader range of cohorts and isolating the effects of other micronutrients within gold kiwifruit implicated in the pathophysiology of depression is warranted. Overall, preliminary evidence suggests that daily consumption of two gold kiwifruit might improve psychological well-being in adult populations.

## 1. Introduction

### 1.1. Mental Health and Psychological Well-Being

The global burden attributable to depressive and anxiety disorders in adults is considerable with depressive and anxiety disorders ranked as the sixth and fifteenth leading cause of disability-adjusted-life-years (21.6 & 28.7 million respectively) in adults aged 25-49 years in 2019 [1,2]. Recognising the lifespan prevalence and the growing burden of depressive disorders, between 1990 and 2019 depressive disorders rose from the nineteenth to the thirteenth leading cause of disability-adjusted-life-years across all ages [3]. The World Health Organisation estimated that in 2015, 4.4% of the global population (322 million) were living with depression [4]. Twelve-month and lifetime prevalence rates of depression in adults have been calculated at 7.2% and 10.8%, respectively, [5]. The psychosocial burden on individuals living with depression is substantial including negative effects on quality of life, activities of daily living, cognition and mood, employment status, mortality and interpersonal relationships [6,7]. The financial cost associated with depression for individuals and society are not inconsequential. In developed nations, health care costs for individuals with depression can be 30% greater than those in the general population [8] and societal costs in the USA were estimated at USD326.2 billion in 2018 [6].

Despite the high prevalence rates and level of disease burden wrought by depression, and seemingly at odds with our knowledge of the condition and efficacious treatments (e.g., pharmacotherapy and psychological interventions), the frequency of help-seeking rates remains concerningly low. Global estimates of the treatment gap for depression, representing the difference between the prevalence of the disorder and the proportion of individuals treated for depression, suggest more than half (56%) of all individuals with depression do not receive any treatment [9]. A review paper examining the World Health Organization’s World Mental Health Surveys reported that 56.7% of individuals with Major Depressive Disorder identified the need for treatment [10]. Of these, 71.1% had attended at least one visit to a service provider, however only 41% received treatment that satisfied minimal standards.

Although depression is diagnosed and treated in health care settings, barriers exist to professional treatment delivery. Access to treatment providers, costs of treatment, and demand for services that outstrip supply contribute to low levels of treatment and high prevalence rates [10]. Fortunately, treatment for depression is not restricted to pharmacotherapy and psychological interventions. Lifestyle factors such as regular exercise, positive social supports, stress management, adaptive sleep hygiene behaviours and healthy dietary patterns have been associated with reduced risk of depression and higher psychological well-being [11,12,13]. Moreover, many of these factors are accessible outside of formal treatment pathways and allow the individual to exercise influence over their psychological well-being. Supporting the positive consumer perception of the value of lifestyle factors in moderating mental health outcomes, a recent national survey identified that almost 61% of respondents took some form of self-initiated action to improve their psychological well-being [14]. These actions included increasing their level of physical activity (37%), accessing social supports (24%) and practising relaxation techniques (23%). Recognising growing public interest in the effect of diet and nutrition on mental health outcomes, more than 20% of respondents reported changing their diet and 15% initiated vitamin and supplement intake to maximise their psychological well-being.

### 1.2. Diet and Psychological Well-Being

A recent randomized controlled trial (HELFIMED trial) examined the influence of a Mediterranean-style diet, characterised by high intake of legumes, nuts, seeds, wholegrains, fruit, vegetables and olive oil, supplemented with fish oil in adults with self-reported depression [15]. Results at three months showed that adherence to the Mediterranean diet was associated with increased consumption of fruit, nuts and vegetables and was associated with reductions in symptoms of depression. A Mediterranean diet intervention was also associated with improved self-reported quality of life and decreased depression symptomatology in young males [16]. Offering another pathway to improved psychological well-being, provision of nutrition education counselling (i.e., emphasising the benefits from fruit and vegetable consumption and biological effects of active constituents such as vitamin C) was associated with increased fruit and vegetable consumption, increases in plasma vitamin C concentrations and improvements in quality of life in a community setting [17].

The consequences of inadequate fruit and vegetable intake on adult psychological well-being are substantial. Population-based surveys have reported that higher fruit and vegetable consumption was associated with reduced odds of experiencing an anxiety or mood disorder, positive mental health, and greater psychological well-being [18,19]. Elsewhere, it has been reported that five serves of fruit and vegetables per day conferred reduced risks of anxiety and cognitive impairment [20,21]. Longitudinal studies have reported that increased fruit and vegetable consumption was associated with reduced psychological distress and risk of depression, and increased well-being, life satisfaction and happiness [22,23,24,25]. Thus, evidence to date suggests that dietary interventions, both those that adopt specific food group or generalised approaches to improved dietary intake, might offer useful adjunctive avenues for the prevention and treatment of common mental health disorders and improve psychological well-being.

### 1.3. Vitamin C and Psychological Well-Being

In addition to considering the impacts of whole of diet and food-group approaches on well-being and mental health, research has also considered the role of specific vitamins and minerals. Vitamin C is one such micronutrient that has garnered significant attention in the diet and well-being landscape. Vitamin C is a water-soluble micronutrient that humans, along with only several other species, are unable to synthesize themselves despite having an absolute requirement for vitamin C for a range of important biological functions [26]. The dialectical opposition between an absolute requirement for vitamin C and our species’ loss of ability to synthesise it, has been explained by the concept that our early ancestors relied on diets rich in vitamin C which led to the eventual pruning of genes involved in endogenous vitamin C synthesis [27]. Therefore, vitamin C must be obtained exclusively from diet, principally through the consumption of fruits and vegetables [28]. Vitamin C acts as an antioxidant and free radical scavenger and is an essential cofactor in numerous enzymatic reactions including that of dopamine *β*-hydroxylase, an enzyme that is central to the synthesis of adrenaline from dopamine. Vitamin C also acts as an essential cofactor in the metabolism of tryptophan, a necessary requirement for the synthesis of serotonin [29,30,31]. Altered dopamine *β*-hydroxylase activity has been described in a range of psychiatric conditions including mood and anxiety disorders and disorders of the digestive tract [32] and acute tryptophan depletion has been associated with reduced serotonin levels and lowered mood states [33]. Emerging work in the field of epigenetics indicates that vitamin C contributes to epigenetic modifications in early development which in turn may influence key psychological and physiological outcomes across the lifespan [34,35]. Reinforcing its role in neurocognitive functioning, the highest concentrations of vitamin C in humans are found in the brain and cerebrospinal fluid and vitamin C is preferentially retained in these areas even when plasma and other organs in the body are depleted of vitamin C [36].

The relationship between vitamin C and psychological well-being in adult populations has been examined through cross-sectional studies, albeit with mixed results. Depression symptoms along with overall mood disturbance, anger and confusion were inversely related to plasma vitamin C concentrations in a cross-sectional study of male tertiary students [31]. In contrast, others have reported no significant associations between fatigue and mood in a similar-aged cohort [37] nor in a cohort of healthy adults aged 50-years old [28]. Heterogeneity of ethnicity may have contributed to the mixed results between these studies [38]. Intervention studies delivering vitamin C in supplement form have also returned mixed results. Acutely hospitalised elderly patients supplemented with 500 mg vitamin C twice-daily for 7–14 days experienced reductions in mood disturbance scores and psychological distress [29,39,40]. Healthy young adults supplemented with 1000 mg/day vitamin C (3 x/day) for 14 days exhibited a reduced subjective stress response [41], and 500 mg/day supplementation of vitamin C was associated with mood improvements and increased vigour in healthy adult males [42]. Conversely, no associations were found between vitamin C supplementation and levels of fatigue or mood in healthy adults supplemented with 1000 mg/day in a recent placebo-controlled trial [37]. Against the backdrop of the established associations between adequate fruit intake, plasma vitamin C concentrations and psychological well-being, the equivocal results from supplement studies compels researchers to investigate whether mental health outcomes might be supported by daily consumption of fruits that are recognised for their high vitamin C concentrations.

### 1.4. Kiwifruit

Kiwifruit, the commercial designation of plants from the genus Actinidia are a nutrient-dense fruit that contain high levels of micronutrients including vitamins, minerals, proteins as well as compounds such as phytonutrients, enzymes and other antioxidants [38]. Importantly, kiwifruit are recognised for their exceptionally high vitamin C content. The two most commercially popular and studied varieties are Actinidia chinensis var. chinensis ‘Zesy002’ Zespri^®^ SunGold and Actinidia deliciosa ‘Hayward’ kiwifruit (hereinafter referred to as gold kiwifruit and green kiwifruit, respectively).

A growing body of research has demonstrated the putative health benefits associated with gold and green kiwifruit consumption. Specifically, both varieties have positive associations with digestive and metabolic health outcomes [43]. More recently, research has begun to link potential benefits of gold kiwifruit consumption to improved psychological well-being. A single serve (90 g) of gold kiwifruit delivers more than three times the recommended daily intake of vitamin C for adults (45 mg/d) and contains almost 70% more vitamin C than the green kiwifruit (137 mg & 79 mg respectively) [44,45]. When judged against other commercially popular fruits, the vitamin C content of gold kiwifruit easily surpasses the levels typically found in oranges (46.9 mg), strawberries (41 mg), bananas (5 mg), blueberries (3.5 mg) and pineapple (22.5 mg) per single serve [45]. The high vitamin C content of gold kiwifruit has it well-placed to serve as an important whole food vector for the delivery of vitamin C, a micronutrient with established associations with psychological well-being.

## 2. Aim of Review

The aim of this rapid review is to summarise the available literature from randomized controlled trials that have examined the impact of kiwifruit interventions on psychological well-being in adult populations

## 3. Method

### 3.1. Research Question

Does kiwifruit consumption support psychological well-being in adult populations through increased vitamin C intake?

### 3.2. Design

This review was conducted in line with the Preferred Reporting Items for Systematic Reviews and Meta Analyses (PRISMA 2020 statement) and PRISMA for abstracts [46,47] with consideration given to applicability of PRISMA 2020 statement items to a rapid review. The rapid review method was selected as the authors identified a priori that a paucity of literature existed pertaining to the research question. Furthermore, it was recognised that this style of review was well suited to introducing primary research articles, informing clinical practise of the development of new treatments, and responding to specific research questions, as was the case in the present article [48,49]. Rapid reviews have been shown to produce similar results compared to systematic reviews, are rooted in the scientific method, and can offer adequate advice for clinical decision-makers [50,51,52].

### 3.3. Inclusion/Exclusion Criteria

Eligibility criteria were determined using the PICO (population, intervention, comparisons and outcome) framework [53] and was arrived at through author consensus. Studies were included if they were randomized controlled trials or observational trials of adult (≥18 years of age) populations. Studies were selected if the active intervention was identified as any form (i.e., whole fruit or kiwifruit-derived nutritional ingredients) of gold or green kiwifruit, or if, in the case of observational studies, kiwifruit was identified as an explanatory variable. Intervention studies that utilised a methodologically justifiable comparator (i.e., placebo or another dietary condition) were included. Studies were excluded if they did not quantify the amount of kiwifruit consumed, delivered kiwifruit as part of a larger intervention (i.e., Mediterranean Dietary Pattern), did not contain original data or if results of the outcome(s) of interest were not measured or reported.

The primary outcome measure of interest was psychological well-being. Psychological well-being was considered a catch-all term to encompass a range of constructs including mood, psychological distress, depression, anxiety, stress, vitality, vigour and mood disturbance. Only studies that assessed psychological well-being with validated and reliable psychometric assessment instruments were considered for inclusion. A secondary outcome measure was improvements in vitamin C concentrations determined by validated assessment tools (e.g., blood plasma pathology protocols). Studies that reported on vitamin C concentrations but did not assess psychological well-being were excluded. Eligibility criteria for inclusion was restricted to articles written in the English language and published between 2005 to the date of the database search. Restricting the search to English-language articles located in major bibliographic databases (Table 1.) can produce results comparable to reviews using more comprehensive searches sans language restrictions and is considered a viable methodological protocol for rapid reviews [54,55].

### 3.4. Databases

Multiple databases (interface/platform) were utilised in the literature search to maximise integrity of the review [48]. CINAHL with full text (EBSCOhost), Embase (Ovid) and PubMed (NCBI) were searched on 5 July 2022.

### 3.5. Search Terms

The key concepts identified in the research question (Kiwifruit, mental health, and vitamin C) were used to generate search terms developed by the first author. Final search terms were arrived at via consensus following review and discussion between MB and co-authors, NK and IZ. Two studies [56,57] previously identified as pertinent to the research question were reviewed for key words to support formulation of search terms. An experienced and qualified librarian was engaged to help develop the search strategy and identify suitable databases and a basic logic grid was created and applied to the individual databases. Search terms (Table 1) remained consistent across databases with adjustments made to reflect differences in subject headings, field codes and truncations used within each database. The three key concepts were combined with AND in the search (i.e., Kiwifruit AND Mental health AND Vitamin C) to ensure that results mentioned all key concepts in the same article. The first author was responsible for undertaking the database search.

### 3.6. Screening

Title, abstract and full text screening of articles were completed by the first author.

### 3.7. Critical Appraisal

The first author was responsible for critical appraisal of the selected studies. Studies were appraised using the Joanna Briggs Institute (JBI) critical appraisal checklist for randomized controlled trials [58].

## 4. Results

### 4.1. Search Outcomes

Twenty-two articles were captured in the initial search of the databases (CINAHL (*n* = 3), Embase (*n* = 11) and PubMed (*n* = 8). The study selection process is summarised in the PRISMA flowchart (Figure 1). The citations were exported and combined into a single library using Endnote 20 software [59]. Full-text documents were retrieved for all articles.

After the removal of duplicates (*n* = 9), thirteen articles remained for title screening. Of these, eight were assessed as irrelevant to the research question and excluded. Five assessed kiwifruit micronutrient profiles or kiwifruit harvesting outcomes [60,61,62,63,64], one addressed chemotaxis and oxidant generation in humans [65], another examined the impact of functional foods on bone health [66], and one article was identified as a study methodology that considered the relationship between kiwifruit and sleep [67]. Three of the remaining five articles were removed following full-text screening. These included an article identified as a commentary on one of the other articles [68], an article providing a summary of studies that examined nutritional intake and neurocognitive health [69], and a cross-sectional study that examined the relationship between blood plasma vitamin C status and psychological well-being without kiwifruit specificity [38]. The remaining two articles [56,57] were assessed as eligible for inclusion and critical appraisal.

### 4.2. Critical Appraisal

The included articles by Carr et al. [57] and Conner et al. [56] satisfied more than half (9 and 8 items, respectively out of 13 items) of the criteria on the JBI appraisal tool and were deemed suitable for inclusion in the review. Both studies failed to satisfy checklist criterion related to the blinding of participants and researchers, an issue regularly encountered in clinical trials delivering whole-food interventions [70].

### 4.3. Description of Included Studies

A summary of the studies included is provided at Table 2.

Carr et al. [57] investigated the potential mood-enhancing properties of gold kiwifruit in young adult males (*N* = 35) with sub-optimal vitamin C concentrations at screening using a parallel-arms design. Participants were randomized to receive either one half of a gold kiwifruit per day or 2 gold kiwifruit per day across a 6-week intervention period following a 5-week lead-in. Psychological health as indicated via mood (65-item POMS) and plasma vitamin C assessment was completed at baseline and at the end of the intervention period. Following the intervention, participants (*n* = 17) supplemented with 2 gold kiwifruit per day displayed a trend towards a 35% decrease in total mood disturbance (*p* = 0.061) and depression (*p* = 0.063) on the POMS. A sub-group analysis of participants (*n* = 8) in the two kiwifruit/d condition with higher-than-average baseline total mood disturbance revealed a 38% decrease in mood disturbance (*p* = 0.029), a 38% decrease in fatigue scores (*p* = 0.048), a 31% increase in vigour (*p* = 0.023) and a 34% decrease in depression trending toward significance (*p* = 0.075). No effect on mood scores was observed in participants (*n* = 9) in the two kiwifruit/d condition with lower-than-average baseline total mood disturbance scores. Analysis of participants (*n* = 18) in the ½ kiwifruit per day condition revealed no effect of the intervention on mood outcomes. Plasma vitamin C concentrations increased significantly in both the low-dose and high-dose conditions following the intervention; however, a 15-fold increase in urinary vitamin C excretion was observed in the high-dose condition suggesting plasma ascorbate saturation for the high-dose group only.

Conner et al. [56] employed a randomized, placebo-controlled trial examining the effects of gold kiwifruit on psychological well-being and subjective vitality. Young adults (*N* = 167, 61% female) aged 18–35 years with sub-optimal plasma vitamin C (<40 µmol/L) were randomized to receive either 2 gold kiwifruit per day, a daily 250 mg vitamin C supplement or a placebo tablet (1 tablet/day) across the 28-day intervention period, book-ended by two-week lead-in and two-week washout periods. Participants completed psychological well-being (POMS-SF, MFSI & WEMWBS) and fasting plasma vitamin C assessments at fortnightly intervals throughout the 8-week trial. Participants who consumed gold kiwifruit reported significant (*p* = 0.03) improvements in mood and well-being during the intervention period; and well-being improvements (*p* = 0.02) persisted during washout. Participants in the supplement condition exhibited non-significant improvements in well-being and decreased fatigue across the intervention period. Participants in the kiwifruit and supplement conditions returned significant improvements in plasma vitamin C. No effects on any outcomes were observed in the placebo condition.

## 5. Discussion

The purpose of this review was to identify and describe studies that have examined the potential role of kiwifruit in supporting psychological health and well-being in adult populations. The review identified a very small but promising body of work that provided preliminary evidence of benefits to psychological well-being from consumption of gold kiwifruit. The search did not reveal any studies that had examined the effect of green kiwifruit on psychological well-being. This is not surprising given the substantial difference in vitamin C concentrations between the green and gold cultivars (79 mg and 137 mg per 90 g serve, respectively) and the recognised links between vitamin C intake and mental health [37,44]. The primary findings reported in the studies identified in the review were that consumption of gold kiwifruit tended to improve overall mood in young adult males with sub-optimal vitamin C concentrations and moderate mood disturbance [57]; and that vitamin C intake via gold kiwifruit improved subjective vitality, and conferred additional benefits compared to a vitamin C supplement in adults with suboptimal vitamin C status [56].

Strengths of the two studies [56,57] in the current review included the use of validated and reliable measures of psychological well-being and plasma vitamin C concentrations, recruiting participants with sub-optimal vitamin C levels, comprehensive screening protocols, lead-in periods to stabilise dietary intake, and intervention periods of sufficient length to assess reliable indices of change from the intervention. The findings in Conner et al. [56] are strengthened by the inclusion of a placebo and comparator group, as well as multiple assessment timepoints across the trial. Nevertheless, the generalisability of results from the two studies is limited by several factors. These include the small sample size (*N* = 35) of young adult males and the small sample size (*n* = 8) included in sub-analyses [57], restricting recruitment to otherwise healthy adults aged 18–35 years old [56,57] and lack of a washout period [57]. Furthermore, micronutrients (i.e., folate, zinc, selenium, magnesium and serotonin) found in gold kiwifruit that have been implicated in the pathophysiology of depression [71,72,73,74] were not isolated or measured by either study, leaving the potential relative contribution to mood outcomes from these nutrients undetermined.

As with most whole food nutrition trials, blinding of participants and researchers to condition allocation is virtually impossible [71] and both studies were not immune to this challenge. One study [56] did incorporate double-blinding, but only between the placebo and comparator conditions and not in the active intervention. Non-blinding issues notwithstanding, and although not directly addressed by Carr et al. [57], the authors may have nullified any placebo effect with the inclusion of the low-dose intervention condition. Delivering an intervention at non-therapeutic levels is an established methodology to isolate the curative agent as the only difference between the groups receiving the intervention [75,76]. Carr et el. [57] recruited participants with sub-optimal vitamin C levels (≤50 µmol/L) and delivered 53 mg of vitamin C to participants in the low-dose condition, well below the estimated intake (83.4 mg) required to achieve optimal plasma concentrations (>50 µmol/L) [77]. Results support the authors’ choice to deliver 53 mg as the non-therapeutic dose with results demonstrating that although vitamin C concentrations increased significantly between baseline and post-intervention in the low-dose condition, participants remained vitamin C-deplete (46 µmol/L) at the end of the trial. Similar, non-therapeutic effects have been reported in other studies delivering 50 mg as a low-dose vitamin C intervention [78]. Others [79] have taken issue with the randomization protocols used by Conner et al. [56] and proposed that a mix of randomisation methods were employed (i.e., stratification and block randomisation), and that some participants were not randomly allocated to groups thereby reducing the strength of causal inferences that might be drawn from results.

Generalisability of results to other populations (e.g., adults 35+ years, non-students, adults with moderate mood disturbance or different ethnicities) remains problematic. Applicability of results to both sexes remain limited as only one of the studies [56] included female participants. Taken together, results from Carr et al. [57] and Conner et al. [56] highlight the potential mood and vitality-enhancing properties of gold kiwifruit and offer a tantalising taste of its utility as a whole-food vector for vitamin C delivery in some adult populations. The differences in effects observed between whole-food and supplement-delivered vitamin C suggest that there are quantifiable, additional benefits from whole fruit over-and-above those conferred by supplements in otherwise healthy adults [56]. Likewise, the enhanced mood and vitality outcomes observed in individuals with greater psychological distress demand further attention and greater individuation of the therapeutic benefits of the panoply of nutrients found in gold kiwifruit.

The scarcity of research exploring the relationships between kiwifruit and psychological well-being offers a range of possibilities for future research. First, given the significant burden of mental illness, it is necessary to replicate the current findings in larger randomized controlled trials involving participants with elevated psychological distress and sub-optimal vitamin C status. Second, opportunity exists to extend research to include child, adolescent, middle-aged and elderly cohorts. Third, given the burden of mental health disorders globally, preliminary evidence of benefits of kiwifruit consumption on psychological health should be extended to populations with clinical or at least sub-clinical symptoms of psychopathology to examine potential benefits of increased kiwifruit consumption, and therefore vitamin C intake, on mental health symptoms and psychological well-being. This is particularly important because the overall influence of kiwifruit on mood outcomes may depend on severity of underlying mental health conditions. Fourth, emerging concepts at the nexus of epigenetics and nutrition such as the influence of diet and nutrient intake in epigenetic regulation and individual differences in response to hypo-nutrition warrant ongoing investigation of the role of essential micronutrients, such as vitamin C, on physiological and psychological outcomes. This would support the development of dietary patterns that could inform individual-specific dietary management strategies to promote life-long psychological well-being.

Whilst Vitamin C has been defined as a putative pathway for the observed benefits of kiwifruit intake, this hypothesis does not preclude other nutritional aspects that may contribute to observed effects. Future studies would benefit from investigating the individual or synergistic contribution to psychological well-being from other micronutrients found at dietary relevant levels in gold kiwifruit with established links to depression pathogenesis [71,72,73,74]. For example, gold kiwifruit contain nutritionally relevant levels of vitamin E, a micronutrient linked to mood regulation [80] and thought to act synergistically with vitamin C in an antioxidant capacity [81]. Furthermore, examination of other minerals and micronutrients endogenous or exogenous to kiwifruit that possess a synergistic relationship with vitamin C is warranted. For example, the bioavailability of iron, a trace element essential for serotonin and dopamine synthesis [82] and implicated in alterations in mood and behavior [83], is enhanced by vitamin C intake. Indeed, preliminary research has already reported that consumption of gold kiwifruit with an iron-fortified meal improved iron status in women with low iron stores [84]. Although not measured in the studies included in this review, gold kiwifruit may confer indirect benefits to psychological well-being through other vectors. The dietary fibre in kiwifruit is recognised as a natural, palatable remedy for improving gastrointestinal function and laxation markers in some populations [85]. Recent research has identified that gold kiwifruit intake increased bowel frequency and reduced markers of gastrointestinal discomfort (e.g., bloating, indigestion) in constipation-compromised adults [86,87]. Given the association between dietary fibre intake and the risk for development of depression [88], the examination of indirect pathways to improving psychological well-being offered by gold kiwifruit may prove fruitful.

The strength of this review is reinforced by utilising multiple databases in the initial search, an extensive and rigorous suite of selection criteria, use of a librarian in the formulation of the search strategy, and three-author consensus of selection criteria and search terms. In line with recommendations of the Cochrane Rapid Reviews Methods Group, restricting articles to English-language only was considered a practical and viable methodology and one which would not necessarily affect overall conclusions [54]. To support review rigour, manual screening of the reference list of the two studies included in the review was also undertaken to detect any missed studies omitted from electronic searching.

Although author consensus determined final search terms, the ensuant articles were screened for title, abstract and full text, and were submitted for critical appraisal by a single-reviewer (MB). Conclusions offered by the review or suggestions for future research are naturally limited by the small sample size and heterogeneity between the studies. Future reviews in this field will be assisted by an increased volume of research. Imposing a restriction to publication date (2005 to July 2022) may have led to some articles being missed [54]. The decision to establish a publication date restriction was taken however, given a priori knowledge that research in this area is in its chronological infancy.

## 6. Conclusions

The aim of this review was to examine the available literature from randomized controlled trials and intervention studies that have investigated the associations between kiwifruit and psychological well-being in adult populations. This review is the first to examine the discrete relationship between kiwifruit, recognised for its exceptionally high vitamin C concentration, and psychological well-being. Results from the two studies concord with previous research that has equivocally established the relationships between vitamin C intake and mental health. Despite the small number of studies included in the review, results warrant further investigations across an expanded range of cohorts and settings to further determine the impact of gold kiwifruit on psychological well-being. Recognising the impact of diet, and in particular whole fruit, on not only physiological, but psychological health, gold kiwifruit may yet take its place within a clinical framework as an appetising option within the dietary intervention space in the prevention and treatment of depression.

## Figures and Tables

**Figure 1 nutrients-14-04657-f001:**
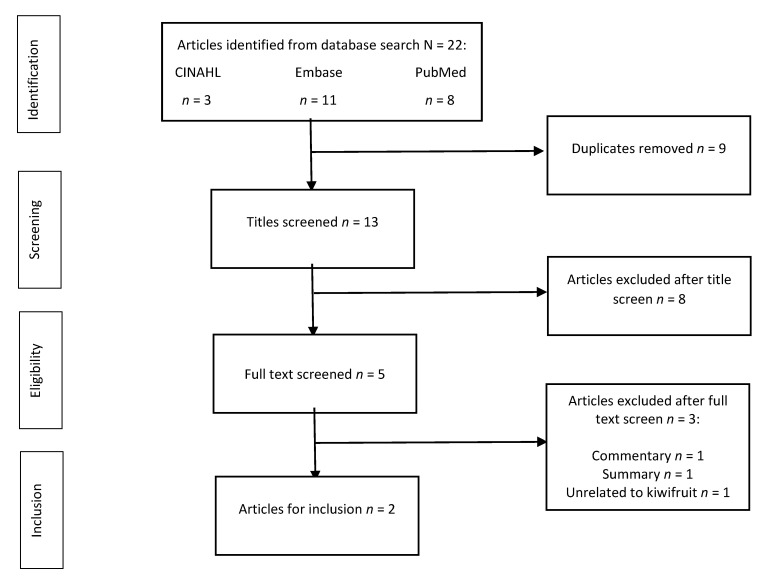
PRISMA flow diagram of rapid review.

**Table 1 nutrients-14-04657-t001:** Search terms (search conducted 5 July 2022).

Key Concept→ ↓Database (Platform)	Kiwifruit	Mental Health	Vitamin C
CINAHL with full text (EBSCOhost)	MH Kiwi + OR TI “Kiwi fruit” OR AB “Kiwi fruit” OR TI Kiwifruit OR AB Kiwifruit OR TI “Actinidia Deliciosa” OR AB “Actinidia Deliciosa” OR TI “Actinidia Chinensis” OR AB “Actinidia Chinensis” OR TI “Gold kiwifruit” OR AB “Gold kiwifruit” OR TI “Green kiwifruit” OR AB “Green kiwifruit” OR TI Zesy002 OR AB Zesy002 OR TI Sungold OR AB Sungold AND	MH “Mental health” OR MH Affect OR MH Depression OR AB Vitality OR TI Vitality OR MH “Psychological well-being” OR TI “Psychological well-being” OR AB “Psychological well-being” OR TI “Psychological well-being” OR AB “Psychological well-being” OR TI “Psychological distress” OR AB “Psychological distress” OR TI “Mood disturbance” OR AB “Mood disturbance” OR TI “Mood improvement” OR AB “Mood improvement” AND	MH “Ascorbic acid” + OR TI “Vitamin C” OR AB “Vitamin C”
Embase (Ovid)	exp Kiwifruit OR Kiwi fruit.ti,ab OR Kiwifruit.ti,ab OR exp Actinidia OR exp Actinidia chinensis OR Actinidia chinensis.ti,ab OR exp Actinidia deliciosa OR Actinidia deliciosa.ti,ab OR Gold kiwifruit.ti,ab OR Green kiwifruit.ti,ab OR Zesy002.ti,ab OR Sungold.ti,ab AND	exp Mental health OR Mental health.ti,ab OR exp Affect OR Affect.ti,ab OR exp Mood OR Mood.ti,ab OR exp Depression OR Depression.ti,ab OR Vitality.ti,ab OR exp Psychological well-being OR exp Psychological well-being OR Psychological well-being.ti,ab OR Psychological well-being.ti,ab OR Psychological distress.ti,ab OR Mood disturbance.ti,ab OR Mood improvement.ti,ab AND	exp Ascorbic acid OR Vitamin C.ti,ab
PubMed (NCBI)	Kiwi fruit[tiab] OR Kiwifruit[tiab] OR “Actinidia” [mh] OR Actinidia deliciosa[tiab] OR Actinidia Chinensis [tiab] OR Gold kiwifruit[tiab] OR Green kiwifruit[tiab] OR Zesy002[tiab] OR Sungold[tiab] AND	“Mental health” [mh] OR Mental health[tiab] OR “Affect” [mh] OR Affect[tiab] OR Mood[mh] OR Mood[tiab] OR “Depression” [mh] OR Depression[tiab] OR Vitality[tiab] OR Psychological well-being[tiab] OR Psychological well-being[tiab] OR Psychological distress[tiab] OR Mood disturbance[tiab] OR Mood improvement[tiab] AND	“Ascorbic acid” [mh] OR Vitamin C [tiab]

**Table 2 nutrients-14-04657-t002:** Summary of included studies.

Citation	Participants	Study Design	Intervention	Study Timeline	Primary Outcome Measures	Main Findings
Carr et al. (2013) [52]	Males (n = 35), Mean age = 21 ± 3 years	Experimental with randomized parallel-arms, no control group	½ kiwifruit/d (n = 18), 2 kiwifruit/d (n = 17)	5-wk lead-in, 6-wk intervention	POMS–TMD; plasma vitamin C	Trend toward decrease in TMD (*p* = 0.06) and depression (*p* = 0.06) in 2 kiwifruit condition. No effect in ½ kiwifruit condition. Participants in 2 kiwifruit condition with higher baseline TMD experienced decrease in TMD (*p* = 0.03), decrease in fatigue (*p* = 0.05), increase in vigour (*p* = 0.02) and trend (*p* = 0.07) toward decrease in depression. Significant increase (*p* < 0.0001) in venous vitamin C in both conditions.
Conner et al. (2020) [51]	Males and females (n = 167, 61% female), Mean age = 21 ± 3 years	Randomized placebo-controlled experimental design with comparator	2 Gold kiwifruit/d (n = 57), 1 Vitamin C supplement/d (n = 56), 1 Placebo tablet/d (n = 54)	2-wk lead-in, 4-wk intervention, 2-wk washout	POMS-SF TMD; MFSI; WEMWBS; plasma vitamin C	Decrease in TMD (*p* = 0.03) in kiwifruit condition. Increase in well-being in kiwifruit (*p* = 0.02) and supplement condition (*p* = 0.09). Decrease in fatigue (*p* = 0.052) at week 2 of intervention in kiwifruit condition. Increase in venous vitamin C (*p* < 0.0001) in kiwifruit and supplement condition(s). No effects observed in placebo condition.

POMS-SF = profile of mood states-short form (35-item), TMD = total mood disturbance, MFSI = multi-dimensional fatigue symptom inventory, WEMWBS = Warwick Edinburgh mental well-being scale, POMS = profile of mood states (65-item).

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
