# Peer review of "The Role of Kiwifruit in Supporting Psychological Well-Being: A Rapid Review of the Literature"

_nutrients, 2022, doi:10.3390/nu14214657_

Round 1

Reviewer 1 Report

The submitted manuscript represents interesting material for a wider audience, bearing in mind that kiwifruit  is a favorite fruit of a large number of people and that it is consumed a lot through food. I believe that the manuscript should be accepted for publication with small additions related to the critical attitude of the author in relation to literary data. In conclusion, I expected the author's critical attitude towards the use of kiwifruit as an important nutrient and source of vitamin C. I also expect the authors to discuss the possible potential reasons that lead to the difference in the use of vitamin C and kiwifruit as its source in the context of other important substances present in the fruit.

Reviewer 2 Report

Hello, 

The current study is very interesting, please note that the topic is original, the literature tackled is recent and straight forward, the text is very easy and smooth to read, we can move easily between the different sections of the paper. The conclusions presented are consistent  with the evidence and arguments presented. 
